

# Dust pollution substantially weakens the impact of ammonia emission reduction on particulate nitrate formation

Hanrui Lang[1], Yunjiang Zhang[1*], Sheng Zhong[2], Yongcai Rao[3], Minfeng Zhou[4], Jian Qiu[5], Jingyi Li[1], Diwen Liu[6], Florian Couvidat[7], Olivier Favez[7], Didier Hauglustaine[8], Xinlei Ge[1]

[1]Collaborative Innovation Center of Atmospheric Environment and Equipment Technology, Jiangsu Key Laboratory of Atmospheric Environment Monitoring and Pollution Control, Nanjing University of Information Science and Technology, Nanjing, China

[2]Jiangsu Environmental Monitoring Center, Nanjing, China

[3]Xuzhou Environmental Monitoring Center of Jiangsu, Xuzhou, China

[4]Suzhou Environmental Monitoring Center of Jiangsu, Suzhou, China

[5]Zhenjiang Environmental Monitoring Center of Jiangsu, Zhenjiang, China

[6]Graduate School of Arts and Science, Columbia University, New York City, USA

[7]Institut National de l'Environnement Industriel et des Risques, Verneuil-en-Halatte, France

[8]Laboratoire des Sciences du Climat et de l'Environnement, CNRS-CEA-UVSQ, Université Paris-Saclay, Gif-sur-Yvette, France

*Correspondence to:* Yunjiang Zhang (yjzhang@nuist.edu.cn)





**Abstract.** Dust emissions significantly influence air quality and contribute to nitrate aerosol pollution by altering aerosol
acidity. Understanding how dust interacts with ammonia emission controls is crucial for managing particulate nitrate
pollution, especially in urban areas. In this study, we conducted field measurements of aerosol components and gases
across three cities in Eastern China during the spring of 2023. By combining an aerosol thermodynamic model with
machine learning, we assessed the contribution of dust to aerosol pH and its impact on nitrate formation. Our results show
that changes in ammonia, both in the gas and particle phases, were the main factors affecting aerosol pH, with dust
particles contributing to about 7% of the total pH variation. During dust events, high concentrations of non-volatile ions
increased aerosol pH, leading to higher nitrate levels in particulate form. Machine learning analysis revealed that extreme
dust storms caused a significant change in aerosol pH, enhancing nitrate partitioning. Further simulations indicated that
while reducing ammonia emissions is effective in lowering nitrate levels under normal conditions, this effect is
significantly reduced in dust-affected environments. Dust particles act as a buffer, reducing the sensitivity of nitrate
formation to ammonia emission reductions. These findings emphasize the need to consider dust pollution when designing
strategies for controlling particulate nitrate levels and highlight the complex interactions between dust and anthropogenic
emissions.



## 1 Introduction

Airborne dust is a major component of atmospheric aerosols, accounting for approximately 75% of the global aerosol mass load (Mahowald et al., 2006; Zhao et al., 2022; Chen et al., 2023c). Dust exerts multiple impacts on air quality (Jickells et al., 2005; Rosenfeld et al., 2001), climate (Huang et al., 2011), and human health (Zhang et al., 2023; Goudie, 2014). It can be broadly categorized into anthropogenic dust and natural dust based on sources and emission mechanisms (Chen et al., 2018; Chen et al., 2023a). Anthropogenic dust originates from human activities, such as construction, agricultural and non-exhaust vehicular emissions (Liu et al., 2021). In contrast, natural dust mainly arises from bare surfaces in arid and semi-arid regions (Shao and Dong, 2006), which cover approximately 30% of the global land area (Soussé-Villa et al., 2024; Xin et al., 2023). Beyond anthropogenic influences, over 300 countries worldwide are affected by natural dust pollution (Kurokawa and Ohara, 2020; Notaro et al., 2015). Dust storms originating in arid regions can be transported over thousands of kilometers, significantly impacting downstream air quality and atmospheric chemistry (Tan et al., 2012; Milousis et al., 2024; Sun et al., 2001).

Dust emissions contain nonvolatile cations (NVCs), such as calcium and magnesium ions, which are alkaline substances that can neutralize acidic aerosol components, such as sulfates, thereby increasing aerosol pH (Wu et al., 2013; Ding et al., 2019). Dust particles also engage in heterogeneous reactions with gaseous nitric acid, buffering acidic species and modulating pH dynamics. Aerosol pH is a critical factor in atmospheric chemical processes, influencing gas-particle partitioning of inorganic aerosols (Guo et al., 2018), secondary organic aerosol (SOA) formation (Xu et al., 2015; Zhang et al., 2017; Nguyen et al., 2014), and metal-catalyzed oxidation reactions (Fang et al., 2017). Regional variations in aerosol pH alter the chemical characteristics of atmospheric pollution, affecting pollutant lifetimes and deposition rates, which in turn have profound implications for ecosystems and public health (Guo et al., 2016). Despite the incorporation of aerosol pH modules in some atmospheric chemistry models, inaccuracies in dust emission inventories can lead to biases in estimated aerosol pH, thereby introducing systematic errors in simulating associated chemical processes, such as nitrate formation.

Nitrate has emerged as a dominant component of fine particulate matter ($PM_{2.5}$) worldwide (e.g., China, Europe, the United States, and India), particularly as sulfate aerosol concentrations decline due to sustained $SO_2$ emission reductions (Weber et al., 2016; Geng et al., 2017; Zhai et al., 2021; Hauglustaine et al., 2014; Beaudor et al., 2024). The reaction between gaseous nitric acid ($HNO_3$) and ammonia ($NH_3$) represents one of the primary pathways for the formation of fine mode nitrate (Stelson and Seinfeld, 1982; Metzger et al., 2002). Nitrate formation plays a critical role in atmospheric



chemistry and the global nitrogen cycle, including reactive nitrogen deposition (Chul H. Song, 2000). The gas-particle
partitioning of HNO₃ and nitrate formation is strongly influenced by aerosol pH (Guo et al., 2018; Shi et al., 2019). When
total ammonia (gaseous and particulate) or NVCs are insufficient to fully neutralize aerosol sulfate, HNO₃ will not
condense on aerosol due to low pH (Nenes et al., 2020; Guo et al., 2017a; Vasilakos et al., 2018; Ding et al., 2019).
However, this conceptual framework may oversimplify the influence of aerosol acidity, as it fails to fully consider the
substantial volatility differences between deliquescent aerosols containing sulfates or NVCs and those dominated by
ammonium or nitrate, both of which are highly sensitive to aerosol pH (Nenes et al., 2020; Nenes et al., 2021). In dust-
polluted environments, however, the abundance of alkaline particles, such as calcium ions, can alter nitrate formation
pathways (Seinfeld et al., 1998; Hrdina et al., 2021; Li et al., 2024). Quantitative insights into how urban dust influences
nitrate formation and its regulation remain nevertheless limited.
East Asia, home to some of the world's major dust source regions, significantly contributes to global atmospheric dust
pollution. Under the influence of Mongolian cyclones, dust particles originating from Mongolia are transported long
distances, affecting air quality and atmospheric processes across East Asia (Fu et al., 2014; Sun et al., 2001; Wang et al.,
2021). The Yangtze River Delta (YRD) is a densely urbanized region in Eastern China, where air quality is influenced by
both natural and local anthropogenic dust sources. This region provides an ideal atmospheric setting to investigate the
impact of dust pollution on urban aerosol acidity and nitrate chemistry. Under these contexts, this study examines changes
in aerosol pH, and nitrate gas–particle partitioning (defined as the gas-particle partitioning of HNO₃ combined to its acid
dissociation) under influence of both anthropogenic and natural dust pollution in spring 2023, focusing on three
representative cities (Xuzhou, Zhenjiang, and Suzhou) in the YRD. The contributions of chemical and meteorological
components to aerosol pH and the effects of dust storms on $\varepsilon(NO_3^-)$ are quantified. By integrating statistical analysis
approaches, we further quantify the contribution of different factors to aerosol pH and $\varepsilon(NO_3^-)$. Sensitivity analyses are
conducted to evaluate the effects of $TNH_x$ ($TNH_x = NH_3 + NH_4^+$), $TNO_3$ ($TNO_3 = HNO_3 + NO_3^-$) and $SO_4^{2-}$ emission
controls on nitrate partitioning across varying dust pollution levels, providing a scientific basis for formulating nitrate
pollution control strategies during dust events.
**2.Data and Methods**
**2.1 Sampling site and instruments**
This study selected three cities in the YRD region, China, that represent a gradient of dust transport effects: Xuzhou



(32.18°N, 119.48°E), Zhenjiang (32.16°N, 119.49°E), and Suzhou (31.29°N, 120.61°E). These cities are distributed along
the north-to-south dust transport pathway, enabling a systematic investigation of the impacts of dust transport, including
gradient variations in particle chemical properties, aerosol acidity (pH), and gas–particle partitioning. The sampling sites
comprehensively reflect the gradient effects of dust across different regions. Observations were conducted at
environmental monitoring stations within each city. These urban monitoring sites, located in mixed residential and
commercial areas, are influenced by multiple sources, including industrial and traffic emissions (Zheng et al., 2021).
Water-soluble inorganic ions in $PM_{2.5}$ (e.g., $NH_4^+$, $Na^+$, $K^+$, $Ca^{2+}$, $Mg^{2+}$, $SO_4^{2-}$, $NO_3^-$, $Cl^-$) and gaseous components ($NH_3$,
$HNO_3$, HCl) were continuously monitored using a Monitor for AeRosols and Gases in ambient Air (MARGA) system
(Trebs et al., 2004; Rumsey et al., 2014). The system exhibited high correlation between cation and anion measurements
(Fig. S1). Throughout the monitoring period, ambient air samples were drawn into the system, where aerosols and gaseous
pollutants were separated. Aerosol particles were collected using a wet sampler, dissolved in water to form sample liquid,
and then analyzed via ion chromatography. For gaseous pollutants, air samples passed through a membrane filter to
remove particles before entering a scrubbing tower, where gas-phase components were dissolved in water to form sample
liquid for ion chromatographic analysis (Rumsey et al., 2014). The MARGA system is equipped with automatic
calibration and cleaning functions, ensuring stability and accuracy during long-term operation. The entire process is
controlled by dedicated software, enabling simultaneous monitoring of multiple components and real-time data output
(Schaap et al., 2004). Meteorological data (temperature and relative humidity) were obtained from corresponding
observation stations, while additional meteorological parameters were sourced from the European Centre for Medium-
Range Weather Forecasts (ECMWF) ERA5 reanalysis dataset (https://cds.climate.copernicus.eu/, last access: November
21, 2023). Reginal $PM_{10}$ data were retrieved from the China National Environmental Monitoring Centre
(https://air.cnemc.cn:18007/, last access: November 21, 2023).
**2.2 Aerosol pH estimation**
Aerosol pH is a particle property that significantly influences aerosol formation, yet it is challenging to measure directly.
Traditional methods, such as ion balance and molar ratio approaches, often fail to provide accurate evaluations of aerosol
pH (Guo et al., 2016; Weber et al., 2016). Currently, the most widely used approaches include the ISORROPIA-II
thermodynamic model (Fountoukis and Nenes, 2007). In this study, we employed the ISORROPIA-II thermodynamic
model to estimate aerosol pH (see Eq. 1) as well as the gas–particle partitioning of water-soluble ions, semi-volatile



compounds, and water content. At low RH, aerosols are unlikely to be in a completely liquid state, and secondary organic
aerosols (SOA) may affect the distribution of semi-volatile compounds due to reduced diffusion within the particles, thus
influencing the predicted pH values; At high RH levels, such as RH > 95%, aerosols may deliquesce, and the exponential
increase in water activity (Wi) can introduce significant uncertainty into the pH values (Guo et al., 2017b; Malm and Day,
2001). To improve the model's accuracy, we applied both the forward mode for metastable aerosols and excluded data
with relative humidity (RH) below 35% or above 95% (Nah et al., 2018; Guo et al., 2015). The equation used to calculate
aerosol pH in ISORROPIA-II is as follows (Liu et al., 2022):
$$pH = -\log_{10}\frac{1000\gamma_{H^+}C_{H^+}}{W_i} \tag{1}$$
In the Eq. (1), $\gamma_{H^+}$ represents the activity coefficient of hydrogen ions, which is generally set to 1 (Liu et al., 2022). $C_{H^+}$
denotes the hydrogen ion concentration in the aerosol aqueous phase, expressed in μg m$^{-3}$. $W_i$ refers to the water content
of the aerosol phase output by ISORROPIA-II (in μg m$^{-3}$). By incorporating these parameters, the ISORROPIA-II model
provides a reliable framework for estimating aerosol pH, allowing for accurate analysis of its variation and impact under
different environmental and pollution scenarios, including those influenced by dust events.
**2.3 The gas–particle partitioning of nitrate (ε(NO₃⁻))**
Nitrate, owing to its volatility, exists in the atmosphere in two primary forms. In the particulate phase, it predominantly
appears as semi-volatile ammonium nitrate. However, where ammonia and NVCs fail to fully neutralize aerosol sulfate,
the formation of semi-volatile ammonium nitrate is inhibited. Under such conditions, nitrate tends to remain in the gaseous
phase as HNO₃, which can subsequently transform into more stable coarse-mode salts, such as Ca(NO₃)₂, over time (Guo
et al., 2017c; Vasilakos et al., 2018; Hrdina et al., 2021). ε(NO₃⁻) defined as the ratio between particle-phase nitrate over
TNO₃ serves as a key indicator of nitrate distribution between its gaseous and particulate phases. Changes in aerosol pH,
influenced by varying meteorological conditions, significantly affect ε(NO₃⁻). This study employs Eq. (2) (Guo et al.,
2018; Nenes et al., 2020) to calculate theoretical values of ε(NO₃⁻) for each observational dataset. The results enable a
detailed analysis of how variations in pH across different ranges influence the gas–particle partitioning of nitrate.
$$\varepsilon(NO_3^-) = \frac{H^*_{HNO_3}W_iRT(0.987\times10^{-14})}{\gamma_{NO_3^-}\gamma_{H^+}10^{-pH}+H^*_{HNO_3}W_iRT(0.987\times10^{-14})} \tag{2}$$
In the equation, $H^*_{HNO_3} = H_{HNO_3}K_{n1}$ (mol² kg⁻² atm⁻¹) represent the product of the Henry's law constant and the acid
dissociation constant for HNO₃. R is the ideal gas constant (J mol⁻¹ K⁻¹), and T is the temperature in Kelvin (K). The



temperature dependence for $H_{HNO_3}$ and $K_{n1}$ can be found in Clegg et al. (1998). pH is calculated using Eq. (1). The factor
$0.987 \times 10^{-14}$ is a unit conversion factor used to convert from atm and µg to SI units. $\gamma_{NO_3^-}$ and $\gamma_{H^+}$ are the activity
coefficients for $NO_3^-$ and $H^+$, respectively. Activity coefficient predicted by ISORROPIA-II are $\gamma_{NO_3^-}\gamma_{H^+}$=0.28, $\gamma_{H^+}$ =1
(Guo et al., 2018; Guo et al., 2017b; Nah et al., 2018). In the standard S-curve, pH varies within a specific range, and this
relationship is influenced by the temperature dependence of the Henry's law constant and the acid dissociation constant.
This model allows for a more accurate estimation of nitrate aerosol behavior under varying environmental conditions.
More detailed information about inputs and outputs for the ISORROPIA-II model can be found in Tables S1 – S3.
**2.4 Multi-site concentration weighted trajectory (CWT)**
The CWT analysis method is widely used to assess the potential origins and transport pathways of air pollutants observed
at receptor sites. By integrating trajectory analysis, this approach provides insights into pollutant sources and their
atmospheric transport dynamics. In this study, we employed the CWT model, coupled with backward trajectories and
multi-site air quality monitoring data, to investigate the potential source regions and long-range transport of the spring
2023 dust storm event observed in Xuzhou, Zhenjiang, and Suzhou. When combined with data from multiple monitoring
sites, the CWT model demonstrates enhanced robustness and reliability (Boichu et al., 2019). Briefly, multi-site CWT
analysis integrates pollutant concentration data from several monitoring stations with the corresponding backward
trajectories to estimate the likely origins of the observed pollutants. Air pollutant concentrations are spatially allocated to
grid cells traversed by air masses, weighted by the residence time within each grid cell. Compared to single-site CWT
analysis, the multi-site approach offers broader spatial coverage, minimizes site-specific biases, and increases the dataset
size, thereby improving the accuracy and spatial resolution of source apportionment, particularly for complex transport
patterns.
In this study, 48-hour backward trajectories at 50 meters above ground level were computed using meteorological data
from the Global Data Assimilation System (GDAS). The CWT analysis was conducted using the Zefir toolkit
implemented in Igor Pro (Petit et al., 2017). This methodology provided a comprehensive assessment of dust transport
and source attribution, facilitating a deeper understanding of dust storm dynamics in the region.
$$CWT_{ij} = \frac{\sum_{l=1}^{n} C_l * \tau_{ij,l}}{\sum_{l=1}^{n} \tau_{ij,l}}$$     (3)
In Eq. (3), $CWT_{ij}$ represents the weighted concentration in the grid at the $i$ row and $j$ column, $C_l$ is the pollutant



concentration corresponding to the $l$ trajectory, and $\tau_{ij,l}$ is the residence time of the trajectory in the *(i,j)* grid. *n* denotes
the total number of all trajectories.
**2.5 Machine learning model**
Aerosol pH and $\varepsilon(NO_3^-)$ exhibit nonlinear responses to multiple influencing factors. In this study, we employed a machine
learning approach to investigate the effects of extreme dust storm conditions on aerosol pH and $\varepsilon(NO_3^-)$. Specifically, we
used the random forest (RF) algorithm to construct regression models tailored to aerosol pH and $\varepsilon(NO_3^-)$ for each city
under investigation. The dataset for the RF regression models was divided into a training set (80%) and a test set (20%).
The training set was utilized to build the models, while the test set was used to validate their performance. The input
predictive features for both aerosol pH and $\varepsilon(NO_3^-)$ models included the water-soluble inorganic chemical composition
of aerosols ($Na^+$, $SO_4^{2-}$, $NH_4^+$, $NO_3^-$, $Cl^-$, $Ca^{2+}$, $K^+$, $Mg^{2+}$), gaseous species ($NH_3$ and $HNO_3$), and meteorological
parameters (T and RH). To evaluate the model performance, we applied 5-fold cross-validation for parameter tuning.
Model performance was evaluated using seven statistical metrics: Mean Absolute Error (MAE), Root Mean Squared Error
(RMSE), Normalized Mean Squared Error (NMSE), Mean Bias (MB), Normalized Mean Bias (NMB), Index of
Agreement (IOA), and the correlation coefficient (R). Detailed definitions and calculations for these metrics are provided
in Supplementary Text 1. This machine learning based approach enabled us to quantify the complex, nonlinear
relationships between aerosol properties, chemical compositions, and meteorological conditions, offering deeper insights
into the drivers of aerosol pH and $\varepsilon(NO_3^-)$ under varying dust pollution scenarios.
In addition, SHapley Additive exPlanations (SHAP), a method derived from the Shapley value concept in game theory,
provides an interpretable framework to explain the predictions of complex machine learning models. SHAP quantifies
the contribution of each input variable to individual predictions, making it a powerful tool for understanding model
behavior (Duan et al., 2024; Lundberg and Lee, 2017). In this study, SHAP values were employed to assess the influence
of various factors on aerosol pH and $\varepsilon(NO_3^-)$ under dust storm and local dust conditions. A positive SHAP value for a
given factor indicates that it contributes positively to the prediction, whereas a negative SHAP value implies a suppressive
or inhibitory impact. This analysis allowed us to disentangle the relative contributions of chemical composition,
meteorological conditions, and other variables to the variations in aerosol properties under different dust scenarios.





**3. Results and Discussion**
**3.1 Observational evidence of anthropogenic and natural dust pollution**
Dust emissions can be classified into anthropogenic and natural sources, with $Ca^{2+}$ and $Mg^{2+}$ commonly used as tracers .
Figure 1 shows the relationship between the concentrations of $Ca^{2+}$ and $Mg^{2+}$ during the observation period from March
to April 2023 across the three cities (Xuzhou, Zhenjiang, and Suzhou). It is evident that the concentrations of $Ca^{2+}$ and
$Mg^{2+}$ exhibit two distinctly different linear slopes, indicating that the different dust origins during this period were
influenced by both long-range transport dust storms and local dust emissions. In particular, during the period from April
11th to 13th, a severe dust storm originating was transmitted from northern regions, first impacting Hohhot, and then
southward to the southern cities of the YRD region. As shown in Fig. 2a, the $PM_{10}$ concentrations in the cities along the
transport path exhibited a distinct gradient, with peak values reaching approximately 1702 µg m$^{-3}$ in Hohhot, 1614 µg
m$^{-3}$ in Xuzhou, 925 µg m$^{-3}$ in Zhenjiang, and 576 µg m$^{-3}$ in Suzhou, respectively. In Xuzhou, the average concentration
of $Ca^{2+}$ increased from $0.47 \pm 0.36$ µg m$^{-3}$ during the local dust period to $2.00 \pm 1.66$ µg m$^{-3}$ during the dust storm period,
marking a fourfold increase. Similarly, the average $Ca^{2+}$ concentration rose from $0.30 \pm 0.23$ µg m$^{-3}$ to $1.69 \pm 1.41$ µg m$^{-3}$
in Zhenjiang, while the concentration increased from $0.35 \pm 0.26$ µg m$^{-3}$ to $0.92 \pm 0.52$ µg m$^{-3}$ in Suzhou.

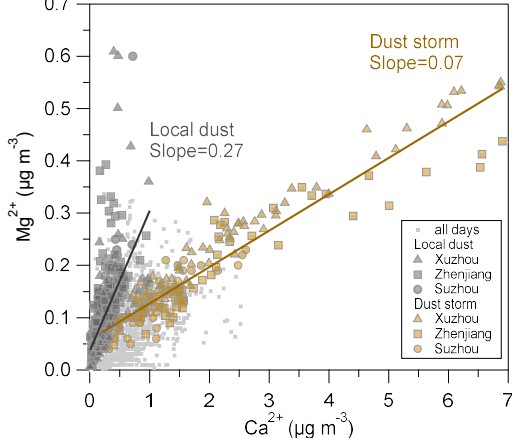


**Figure 1.** Relationship between $Ca^{2+}$ and $Mg^{2+}$ concentrations in $PM_{2.5}$ in Xuzhou (triangle), Zhenjiang (square), and Suzhou (circle).
Dust types are distinguished based on the slope of the $Ca^{2+}$ to $Mg^{2+}$ concentration ratio, with local dust (gray) and dust storm (brown)
indicated. Light gray dots represent the concentrations of $Ca^{2+}$ and $Mg^{2+}$ observed in the three cities during March – April 2023.

Figure 2a and b illustrate the temporal evolution of $PM_{10}$ and $Ca^{2+}$ concentrations during the dust storm, showing an initial



spike in Hohhot, followed by a gradual increase across the Beijing-Tianjin-Hebei (BTH) region, and eventual dispersion
into several cities in Jiangsu Province. This progression is consistent with the CWT-weighted trajectory patterns shown
in Fig. 2c and d, which delineate the transport pathways of the dust storm. The maps highlight significant contributions
from Mongolia – the dust storm's origin – to regions including Hohhot, Beijing, Tianjin, Shijiazhuang, Jinan, Zhengzhou,
and Jiangsu. This finding corroborates the results of Chen et al. (2023b), who attributed the dust storm to a strong cold
high-pressure system and cold front that transported substantial quantities of coarse dust aerosols southward into the YRD
region. Southward-moving cold fronts play a critical role in the diffusion and transport of atmospheric pollutants. In arid
and semi-arid regions, these storms mobilize large amounts of crustal elements, such as $Ca^{2+}$, with high winds lifting dust
from surface sources, including city streets, construction sites, and other exposed land areas (Ding et al., 2019).

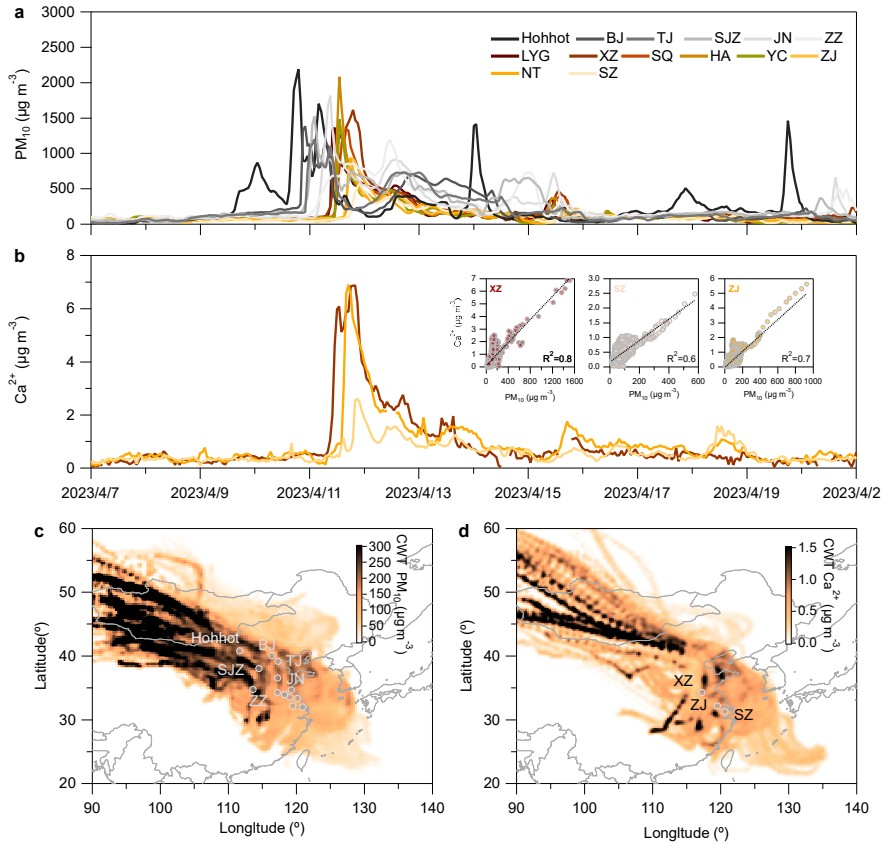

**Figure 2.** Time series of $PM_{10}$ and $Ca^{2+}$ concentrations, and their concentration-weighted trajectories for cities along the dust transport
path. **(a)** Time series of $PM_{10}$ in 14 cities along the BTH region, and **(b)** Time series of $Ca^{2+}$ concentrations in Xuzhou, Zhenjiang, and
Suzhou and the correlation of $Ca^{2+}$ and $PM_{10}$. **(c)** 48-hour CWT-weighted spatial distribution of $PM_{10}$ concentrations in 14 cities from



April 5 to 20, and **(d)** 48-hour CWT-weighted spatial distribution of $Ca^{2+}$ concentrations in Xuzhou, Zhenjiang, and Suzhou (units: μg
$m^{-3}$).

Figure 3 presents the relative contributions within $PM_{2.5}$ water-soluble inorganic species (WSIS) during local dust and
dust storm periods in Xuzhou, Zhenjiang, and Suzhou. Across the three cities, the combined contribution of sulfate, nitrate,
and ammonium consistently exceeded 80% of WSIS, confirming the importance of secondary inorganic aerosols in fine
particulate pollution. Nitrate was the most significant contributor to WSIS during both periods, particularly during the
local dust period, with an average contribution ranging from 49.3% to 52.6%. However, its relative contribution decreased
during the dust storm period, dropping to 34.0% to 40.8%. In contrast, the relative contribution of sulfate increased during
the dust storm period, with increments of 5.2%, 5.0%, and 6.7% in Xuzhou, Zhenjiang, and Suzhou, respectively. This
suggests that the atmospheric dilution and dispersion effects during dust storms might impact nitrate aerosols more
significantly than sulfate. The conclusion of Wang et al. (2022) also supports this result. Indeed, in eastern China, sulfate
aerosols are more regionally distributed as secondary aerosol components, while nitrate formation is more influenced by
local conditions (Wang et al., 2016; Zhang et al., 2015). As expected, during the dust storm period, the relative contribution
of $Ca^{2+}$ and $Mg^{2+}$ increased across all three cities, with an average rise of approximately 10% compared to the local dust
period..

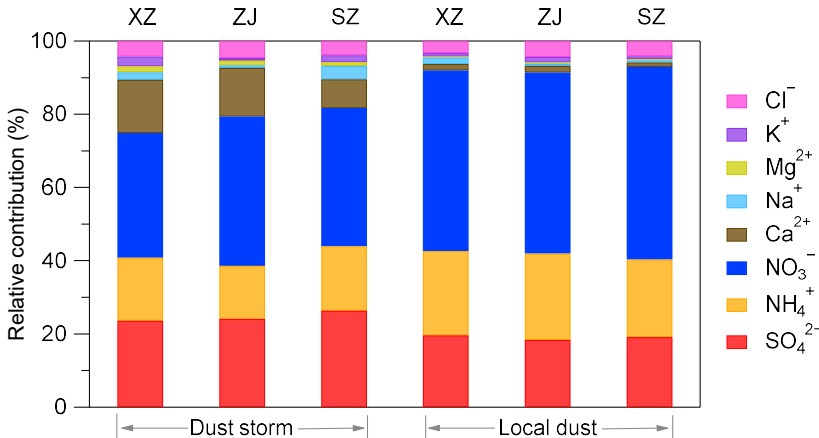


**Figure 3.** Relative contributions of water-soluble inorganics ($SO_4^{2-}$, $NH_4^+$, $NO_3^-$, $Ca^{2+}$, $Na^+$, $Mg^{2+}$, $K^+$, and $Cl^-$) within the $PM_{2.5}$ fraction
in Xuzhou, Zhenjiang, and Suzhou during dust storm and local dust pollution periods, respectively.




### 3.2 Driving factors of aerosol pH

Aerosol pH plays a crucial role in influencing aerosol formation and chemical composition. By regulating the partitioning
of semi-volatile compounds between the gas and particle phases, aerosol pH directly affects the distribution of particulate
matter in the atmosphere (Guo et al., 2017b). To examine the factors influencing aerosol pH, we utilized the ISORROPIA-
II thermodynamic model and sensitivity analysis to quantify the relative contributions of chemical and meteorological
factors, such as T and RH, in Xuzhou, Zhenjiang, and Suzhou. The correlation between simulated and observed
concentrations of $NH_3$ and particulate $NO_3^-$ is presented in Fig. 4. Across all three cities, the simulated values exhibit
strong agreement with measurements ($R^2 = 0.94 – 0.99$). Additionally, Fig. S2 shows high correlations ($R^2 = 0.90 – 0.97$)
for particle-phase ammonium and chloride between ISORROPIA-II predictions and observations, confirming the robust
performance of the thermodynamic model in this study.
To assess the impact of individual factors ($TNO_3$, $TNH_x$, $Ca^{2+}$, $SO_4^{2-}$, T and RH) on aerosol pH, we estimated their relative
contributions using methods like those proposed by Zheng et al. (2020) and Zheng et al. (2022). First, we calculated the
monthly average values for each factor in March and April, referred to as $pH_{i_{(3,3)}}$ and $pH_{i_{(4,4)}}$, respectively. Here, $pH_i$
represents the influence of factor i on pH, with the numbers in parentheses indicating the respective months. For the
analysis of a specific factor, we used the March average value of that factor while holding the other variables at their
average levels for April. This yielded the aerosol pH value, denoted as $pH_{i_{(3,4)}}$. Similarly, when using the April average
value of the factor and maintaining the other variables at their March average levels, we recorded the resulting pH as
$pH_{i_{(4,3)}}$. The relative change in pH, denoted as $\Delta pH_{i_{(3)}}$ and $\Delta pH_{i_{(4)}}$ was calculated as the mean difference between
$pH_{i_{(3,3)}}$ and $pH_{i_{(4,3)}}$, and between $pH_{i_{(4,4)}}$ and $pH_{i_{(3,4)}}$, respectively (see Eqs. 4 and 5). Finally, the overall impact of
each factor on aerosol pH could be estimated (see Eq. 6).
$$\Delta pH_{i_{(3)}} = pH_{i_{(3,3)}} - pH_{i_{(4,3)}} \tag{4}$$
$$\Delta pH_{i_{(4)}} = pH_{i_{(4,4)}} - pH_{i_{(3,4)}} \tag{5}$$
$$\Delta pH_i = \frac{\left[\Delta pH_{i_{(3)}}\right] + \left[\Delta p_{i_{(4)}}\right]}{2} \tag{6}$$
The impact of each factor could be positive or negative, which was detailed in Fig. S3. As shown in Fig. 5, atmospheric
total ammonia emerged as the most significant driver of aerosol pH changes in all three cities, contributing 42%, 57%,

and 43% of the observed pH in Xuzhou, Zhenjiang, and Suzhou, respectively. Total ammonia led to $\Delta pH_{TNH_x}$ increases of 0.6, 1.3, and 0.5 units in these cities during spring 2023. For Zhenjiang, T and $Ca^{2+}$ were the next most influential factors, contributing 0.6 and 0.15 units to $\Delta pH_T$ and $\Delta pH_{Ca^{2+}}$, respectively. Sulfate exhibited the smallest influence on aerosol pH, where a concentration change of 0.3 μg $m^{-3}$ corresponded to a $\Delta pH_{SO_4^{2-}}$ of approximately 0.05 units. These results align with the findings of Weber et al. (2016), which suggest that aerosol pH is less sensitive to changes in sulfate concentrations compared to ammonia levels.

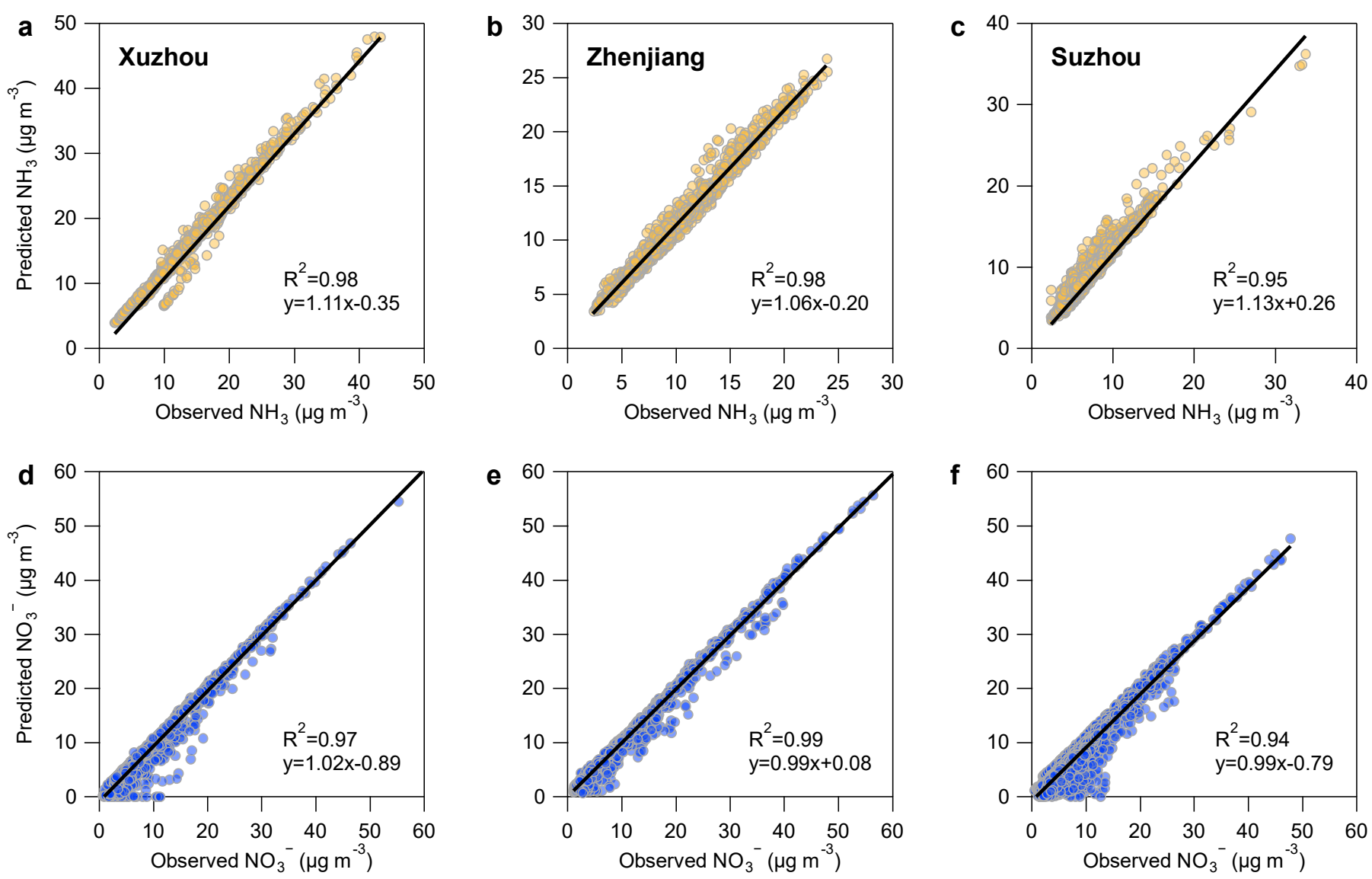

**Figure 4.** Correlation between ISORROPIA-II simulated and observed values of $NH_3$ and $NO_3^-$ in three cities. **(a) – (c)** show the correlation between $NH_3$ predictions and observations, while **(d) – (f)** show the correlation between $NO_3^-$ predictions and observations. The first column represents Xuzhou, the second column represents Zhenjiang, and the third column represents Suzhou.

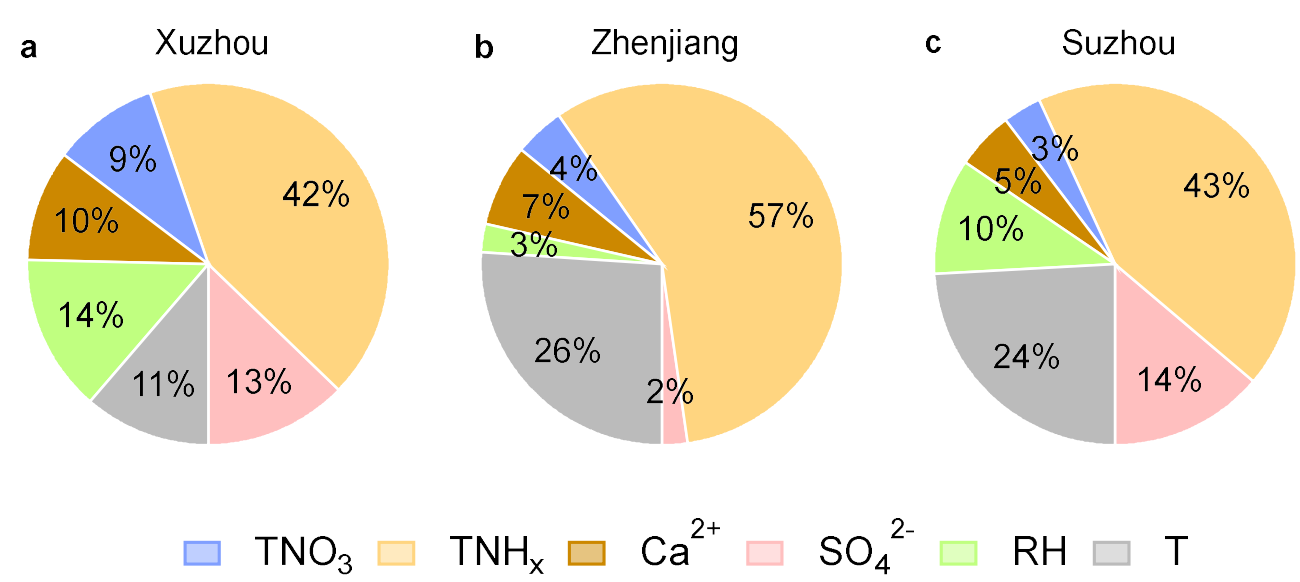

**Figure 5.** Relative contribution of different factors, including $TNO_3 = HNO_3 + NO_3^-$, $TNH_x = NH_3 + NH_4^+$, $Ca^{2+}$, $SO_4^{2-}$, RH, and T, to aerosol pH during the entire observation period in **(a)** Xuzhou, **(b)** Zhenjiang, and **(c)** Suzhou.




To further explore the response of aerosol pH to variations in $SO_4^{2-}$ and $NH_3$ concentrations under different dust conditions
(non-dust, local dust, and extremely dust storm), we conducted sensitivity simulations constrained by observations from
Zhenjiang as a case study. As illustrated in Fig. 6a – c, we extended the $NH_3$ and $SO_4^{2-}$ concentration ranges beyond their
observed values to encompass potential variations across the YRD region. The input concentrations of $Na^+$, $SO_4^{2-}$, total
chloride ($TCl_x = Cl^- + HCl$), $K^+$, and $Mg^{2+}$ were fixed at the average levels observed in Zhenjiang during the study period
(see Table S2). Simulations were carried out under three distinct $Ca^{2+}$ concentration scenarios: (1) non-dust ($Ca^{2+} = 0$ μg
$m^{-3}$), (2) local dust ($Ca^{2+} = 0.7$ μg $m^{-3}$), and (3) extremely dust storm ($Ca^{2+} = 3.00$ μg $m^{-3}$). In these simulations, total
ammonia ($TNH_x = NH_4^+ + NH_3$) and total nitrate ($TNO_3 = NO_3^- + HNO_3$) concentrations were independently changed
and input into the ISORROPIA-II model. Under non-dust conditions ($Ca^{2+} = 0$ μg $m^{-3}$), the model predicted lower aerosol
pH values. As shown in Fig. 6a – b, a 5 – 10-fold increase in $NH_3$ concentration led to a pH increase of approximately 1
unit, whereas aerosol pH demonstrated limited sensitivity to $SO_4^{2-}$ concentration changes. This finding is consistent with
previous studies (Zheng et al., 2022; Weber et al., 2016; Xie et al., 2020). However, under high $Ca^{2+}$ concentration
conditions, such as during extremely dust storm events, the influence of $NH_3$ on aerosol pH was notably mitigated (Fig.
6c). The presence of $Ca^{2+}$ during dust storms reduced the ability of $NH_3$ to modulate aerosol pH effectively. These results
suggest that elevated $Ca^{2+}$ concentrations, a characteristic of dust events, play a significant role in buffering the impact of
$NH_3$ on aerosol pH.

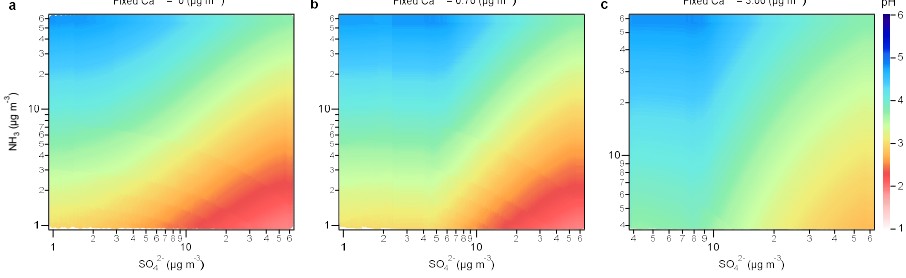


**Figure 6.** Sensitivity of the pH to ammonia ($NH_3$) and sulfate ($SO_4^{2-}$) concentrations based on ISORROPIA-II model predictions under
different $Ca^{2+}$ concentration conditions: **(a)** 0, **(b)** 0.70, and **(c)** 3.00 μg $m^{-3}$.

**3.3 Impact of aerosol pH on the partitioning of nitric acid**
In eastern China, nitrate has become a key component of $PM_{2.5}$, instead of sulfate (Xu et al., 2023; Gao et al., 2023). As



a semi-volatile compound, nitrate is strongly influenced by the gas-particle partitioning process in the atmosphere.
Aerosol pH not only determines the stability of nitrate but also governs whether it exists in the particulate phase or
volatilizes as $HNO_3$ in the gas phase (Guo et al., 2018). At higher pH, nitrate tends to exist in the particle phase due to the
oxidation of $NO_x$, while under lower pH conditions, nitrate is more likely to volatilize into the gas phase as $HNO_3$ (Nenes
et al., 2020). Using Eq. (2), we analyzed the relationship between the nitrate particle-phase fraction ($\varepsilon(NO_3^-)$) and aerosol
pH for three cities – Xuzhou, Zhenjiang, and Suzhou – under dust storm and local dust conditions. Fig. 7 shows the S-
shaped curve representing this relationship, calculated based on the average $T$ and aerosol $W_i$ during dust storm and local
dust conditions, assuming ideal solution behavior (activity coefficient $\gamma_{H^+}$ = 1). This curve visually demonstrates the
regulation of nitrate phase partitioning by aerosol pH under these conditions and provides a theoretical basis for
controlling the effect of ammonia on particulate nitrate formation by adjusting aerosol pH (Guo et al., 2018).
As cities along the dust storm transport path, Xuzhou, Zhenjiang, and Suzhou experience varying degrees of dust influence,
leading to significant differences in aerosol pH. On average, aerosol pH is elevated during dust storms compared to local
dust conditions. During dust storms, the mean aerosol pH values were 5.50 ± 1.65 in Xuzhou, 5.44 ± 1.69 in Zhenjiang,
and 5.30 ± 1.67 in Suzhou. Under local dust conditions, these values were lower, at 4.12 ± 0.52, 3.92 ± 0.32, and 3.74 ±
0.69 respectively. Xuzhou, situated at the northern edge of the dust storm transport path, exhibited the highest aerosol pH
during both periods, reflecting the substantial impact of transported dust pollution. The S-shaped curve in Fig. 7
demonstrates that under both dust storm and local dust conditions, the average aerosol pH aligns with nitrate particle-
phase fractions exceeding 99%, indicating that nitrate predominantly resides in the particle phase. This finding highlights
the promoting effect of dust pollution on the gas-to-particle transformation of nitrate.
When aerosol pH drops below 3, however, $\varepsilon(NO_3^-)$ decreases sharply, signifying the onset of nitrate volatilization into
the gas phase. Notably, when aerosol pH lies in the range of 1 to 3, $\varepsilon(NO_3^-)$ exhibits heightened sensitivity to aerosol pH
changes. This trend was consistently observed across all three cities. Reducing $NH_3$ concentrations is particularly effective
in influencing nitrate gas-particle partitioning when aerosol pH is within this sensitive range, offering a promising strategy
to mitigate regional particulate nitrate pollution. However, environments with dust pollution may disrupt this theoretical
relationship. NVCs (such as $Ca^{2+}$) in dust can neutralize acidic aerosol components, maintaining aerosol pH at relatively
high levels (e.g., pH > approximately 3.5) (Fig. 7). This neutralization effect limits the ability to lower particulate nitrate
concentrations solely by reducing $NH_3$ emissions, necessitating alternative approaches to address nitrate-driven air quality
challenges in dust-influenced regions.

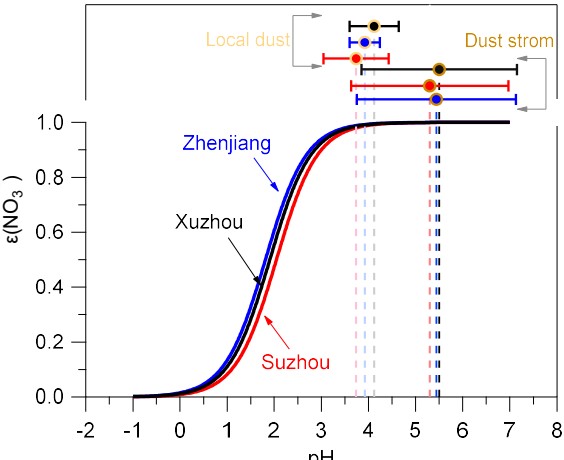


**Figure 7.** S-curve distributions for ε(NO₃⁻) under the conditions from different cities. Based on Eq. (2), the relationship between ε(NO₃⁻)
and pH was calculated using the average T and $W_i$ during dust storm and local dust periods (assuming $\gamma_{NO_3^-}\gamma_{H^+}$=0.28, $\gamma_{H^+}$ =1). The
vertical dashed lines represent the average pH values calculated using ISORROPIA-II for the three cities. Error bars indicate the sample
standard deviation of aerosol pH.

To further quantify the impact of dust storms on aerosol pH and ε(NO₃⁻), we utilized the RF model combined with SHAP
values for both prediction and sensitivity analysis. The correlation between the observed and predicted results from the
RF model is shown in Fig. S4. The Index of Agreement (IOA) values ranged from 0.93 to 0.97, indicating a high level of
model agreement. Meanwhile, the correlation coefficients (R) varied between 0.78 and 0.90, further validating the model's
predictive accuracy. For aerosol pH predictions, five evaluation metrics were used: MAE, RMSE, NMSE, MB, and NMB.
The values for MAE ranged from 0.13 to 0.18, while RMSE values were between 0.26 and 0.29. For NMSE, the values
ranged from 0.10 to 0.12, and the biases (MB and NMB) varied from -0.01 to -0.006 and 0.004 to 0.007, respectively. In
comparison, the corresponding evaluation metrics for ε(NO₃⁻) were as follows: MAE ranged from 0.01 to 0.02, RMSE
from 0.03 to 0.04, and NMSE from 0.10 to 0.21. The bias values for ε(NO₃⁻) ranged from -0.00006 to 0.004 for MB and
from 0.003 to 0.007 for NMB. These statistical results demonstrate the reliability and robustness of the RF model in
predicting aerosol pH and nitrate partitioning.
Figure 8 illustrates the impact of dust storms and local dust conditions on aerosol pH and ε(NO₃⁻). The ΔSHAP values
represent the difference between the average SHAP values of all variables during dust storm periods and the average
SHAP values for all variables during the non-dust storm period. During dust storm conditions, ΔSHAP significantly
increased in Xuzhou, Zhenjiang, and Suzhou, with aerosol pH values rising by Δ1.2, Δ1.5, and Δ1.5 units, respectively





(Fig. 8 a-c). This result is consistent with our previous conclusion that dust storms contribute to an increase in aerosol pH,
confirming the positive impact of dust storms on the random forest model's predictions of aerosol pH. Similarly, Fig. 8
d-f shows the changes in $\varepsilon(NO_3^-)$ for the three cities under different weather conditions. It is evident that the effect of dust
storms on $\varepsilon(NO_3^-)$ is 10 to 20 times greater than the impact of local non-dust storm conditions. This indicates that dust
storm conditions have a significantly stronger positive contribution to the particle–phase fraction of nitrate. The presence
of dust particles facilitates the conversion of nitrate to the particulate phase, highlighting the significant influence of dust
storms on nitrate partitioning in the atmosphere.

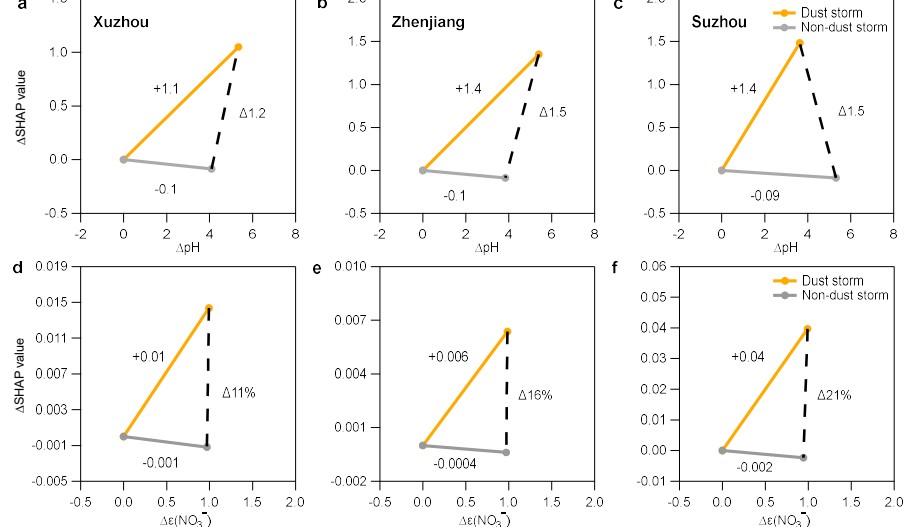


**Figure 8.** ΔSHAP values for **(a) – (c)** aerosol pH and **(d) – (f)** $\varepsilon(NO_3^-)$. The orange solid line represents the impact of dust storms, the
gray solid line represents the non-dust scenario, and the black dashed line shows the difference between the two scenarios.

**3.4 Effectiveness of emission reduction on particulate nitrate under dust pollution**
To explore the impact of emission reductions of $TNH_x$, $TNO_3$, and $SO_4^{2-}$ on $\varepsilon(NO_3^-)$ during different dust storm conditions,
we conducted a sensitivity analysis using the average pollutant concentrations observed in Zhenjiang during the spring of
2023. The results, shown in Fig. 9, demonstrate a nonlinear response of both $\varepsilon(NO_3^-)$ and the total ammonium-nitrate
concentration ($NH_4^+ + NO_3^-$) to reductions in $TNH_x$, $TNO_3$, and $SO_4^{2-}$, respectively. We simulated the effects of
progressively reducing $TNH_x$, $TNO_3$, and $SO_4^{2-}$ by 0% to 50% under different $Ca^{2+}$ concentration conditions, which
include different dust pollution scenarios. For the simulation, $Ca^{2+}$ concentration was set to 0.1 μg m$^{-3}$ for local dust



conditions and ranged from 0.7 to 3.0 µg m⁻³ for dust storm conditions. When the $Ca^{2+}$ concentration exceeded 3 µg m⁻³,
further reductions in the other variables had negligible effects on the output, with emission reductions having little to no
impact on $\varepsilon(NO_3^-)$.
As shown in Fig. 9a, it is evident that during local dust conditions, $\varepsilon(NO_3^-)$ remained relatively constant until $TNH_x$
emissions were reduced by 30%. At this point, $\varepsilon(NO_3^-)$ rapidly dropped from 99%, signaling the onset of a significant
shift in the gas-particle partitioning of nitrate. When $TNH_x$ reductions reached 50%, $\varepsilon(NO_3^-)$ fell sharply to approximately
30%, indicating that nitrate transitioned predominantly into its gas–phase form. This simulation result is consistent with
the sensitivity analysis of $NH_3$ concentrations in section 3.2, which also showed a significant response in nitrate
partitioning as $NH_3$ concentrations decreased. Thus, in the Zhenjiang region, a 30% reduction in $TNH_x$ emissions is
necessary to effectively reduce the mass of $(NH_4^+ + NO_3^-)$ during spring (Fig. 10 d). In contrast, during dust storm
conditions (Fig. 9a), the reduction in $TNH_x$ had a much more subdued effect on $\varepsilon(NO_3^-)$, especially at higher $Ca^{2+}$
concentrations (above 2.5 µg m⁻³), where the reduction had almost no impact on $\varepsilon(NO_3^-)$.
For $TNO_3$ reductions, as shown in Fig. 10 b, the changes in $\varepsilon(NO_3^-)$ were minimal, regardless of the $Ca^{2+}$ concentration.
However, during local dust conditions (Fig. 9e), the reduction of $TNO_3$ led to a significant decrease in $(NH_4^+ + NO_3^-)$
concentrations, indicating that $TNO_3$ reduction was particularly effective under local dust conditions. Lastly, reductions
in $SO_4^{2-}$ emissions (Fig. 9c and f) had a smaller impact on both $\varepsilon(NO_3^-)$ and $(NH_4^+ + NO_3^-)$ concentrations. Interestingly,
at very low dust concentrations, $SO_4^{2-}$ reductions could even lead to a slight increase (by up to 0.5%) in $\varepsilon(NO_3^-)$, indicating
that sulfate reduction alone is not an effective strategy for controlling nitrate partitioning.

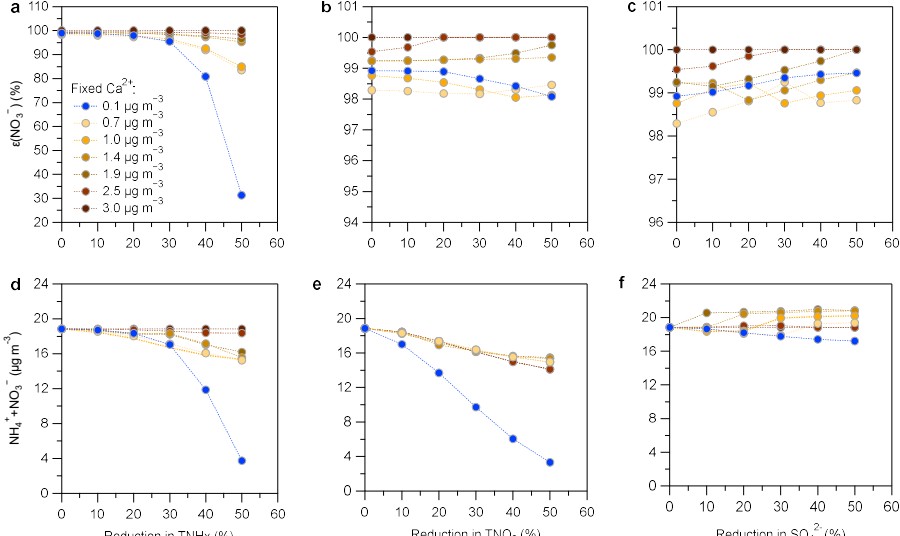

**Figure 9.** Sensitivity analysis simulating the impact of reducing $TNH_x$ ($TNH_x = NH_3 + NH_4^+$), $TNO_3$ ($TNO_3 = HNO_3 + NO_3^-$), and $SO_4^{2-}$ by 0-50% during dust events of varying intensities on $\varepsilon(NO_3^-)$ and $NH_4^+ + NO_3^-$.

## 4. Conclusions and Impactions

This study explores the impact of dust pollution on aerosol pH and nitrate gas-particle partitioning in three cities across the YRD region of Eastern China. By combining field observations, thermodynamic modeling, and machine learning techniques, we provide a comprehensive analysis of how different dust scenarios affect urban aerosol pH and gas-particle partitioning chemistry of nitrate. Our analysis of a dust storm event that originated in Mongolia and was transported over long distances to the YRD region in the spring of 2023 revealed a significant increase in $PM_{10}$ concentrations, the average $PM_{10}$ concentration in three cities along the route exceeds 400 μg m$^{-3}$, approximately four times higher than during local dust events. Thermodynamic simulations using the ISORROPIA model showed that both ammonia and calcium ion concentrations strongly influenced aerosol pH, with average contributions of 47% and 7% respectively. Random forest model simulations further indicated that the presence of high NVCs during dust storms significantly contributed to changes in aerosol pH (1.2 – 1.5 units). Sensitivity analysis of pH responses to sulfate and NH$_3$ concentrations under different dust conditions (non-dust, local dust, and extremely dust storm) revealed that a 5 to 10 fold increase in NH$_3$ led to a 1-unit change in aerosol pH. Machine learning analysis showed that extreme dust storm events contributed



approximately 1.4 units to the increase in aerosol pH, with a corresponding increase in nitrate partitioning (16%). This
suggests that under high aerosol pH conditions during dust pollution periods, nitrate is predominantly in the particulate
phase, indicating that dust significantly inhibits the partitioning of nitrate into the gaseous phase. In addition, our
sensitivity analyses also showed that ammonia reduction had the most significant effect on reducing nitrate aerosols under
dust-free conditions. However, the effectiveness of ammonia reductions in lowering nitrate aerosol concentrations was
significantly reduced due to the influence of NVCs on nitrate partitioning under dust pollution scenarios. These findings
suggest that dust pollution can substantially weaken the impact of ammonia reductions on nitrate aerosol formation,
highlighting the need for targeted control strategies during dust storm events. Dust emission remains a significant air
pollution concern worldwide, while urban nitrate aerosol pollution is a pressing issue in many cities, particularly in East
Asia, where the frequency of natural dust events has increased in recent years. These dust storms, along with
anthropogenic dust, can substantially alter aerosol chemistry by modifying aerosol pH and nitrate partitioning. Therefore,
effective dust control strategies are critical for mitigating the adverse effects of aerosol acidity on nitrate aerosol formation
and improving air quality in dust-prone regions.



*Data availability.* Additional meteorological parameters can be accessed at the European Centre for Medium-Range
Weather Forecasts (ECMWF) ERA5 reanalysis dataset (https://cds.climate.copernicus.eu/; last access: 21 November
2023). Reginal $PM_{10}$ data can be accessed at the China National Environmental Monitoring Centre
(https://air.cnemc.cn:18007/; 21 last access: November, 2023). The additional data will be made available upon request
(yjzhang@nuist.edu.cn).

*Author contributions.*
YZ conceived and designed the study. HL and YZ conducted the simulations and data analysis. HL, YZ, SZ, YR, JQ,
and MZ carried out field measurements and validated the data. HL and YZ wrote the original manuscript, while DL, FC,
OF, HD, and XG provided critical feedback and contributed to the manuscript revisions.

*Competing interests.* The authors declare that they have no conflict of interest.

*Acknowledgements.*
This study was supported by the National Natural Science Foundation of China (grant no. 42207124) and Natural
Science Foundation of Jiangsu Province (grant no. BK20210663).

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
