# Peer review of "emission reduction on particulate nitrate formation"

_EGUsphere, 2025_

## Author Response (AR1)

Dear Editor,

Thank you very much for your and reviewers' thoughtful and constructive comments. We have responded each comment point-by-point and revised the manuscript accordingly. The detailed responses are shown in *blue*, with corresponding changes in the revised manuscript *italicized* in this response letter. The tracked changes in the revised manuscript are marked in *red*.

Thank you very much for your time and consideration.

We are looking for forward to hearing from you.

Sincerely,

Dr. Yunjiang Zhang, on behalf of all co-authors

Nanjing University of Information Science and Technology, Nanjing, China

Email address: yjzhang@nuist.edu.cn  Tel.: 86-(0)25-58731090

**Reply to Anonymous Referee #1**

Comments: This study conducted field measurements of atmospheric aerosol and gaseous species across three urban sites in Eastern China, by employing an integrated approach combining aerosol thermodynamic modeling with machine learning techniques to evaluate the role of dust in modulating aerosol pH and its subsequent effects on nitrate formation. The findings demonstrate that dust composition and ammonia variability constitutes the dominant control on aerosol pH. During dust storm, elevated concentrations of non-volatile cations significantly enhanced aerosol alkalinity, thereby promoting particulate nitrate formation. These processes simultaneously diminished the sensitivity of the aerosol pH to ammonia emission reductions. Overall, the manuscript is well written and the results are valuable to the literature. While the conclusions provide valuable insights, several aspects warrant further clarification.

**Reply:** We thank the reviewer for his/her positive and constructive comments on this manuscript. We have modified the related content according to the comments.

1) Lines 98-99: Please specify the effective particle size capture range of the wet sample, and the corresponding collection efficiency.

**Reply:** Thank you for the comments. We have added an explanation of the wet sample sampling section of the MARGA instrument, including the effective particle size capture range and the corresponding collection efficiency. The revised description reads as follows: *"During the observation period, ambient air was continuously drawn into the MARGA system, where aerosols and gas-phase species were separated. Water-soluble gases were first removed using a wet rotating denuder. Subsequently, aerosol particles with aerodynamic diameters ⩽2.5 µm were collected using a wet aerosol sampler, where they were dissolved in water to form a sample liquid that was then analyzed by ion chromatography."*

2) Lines 233-236: Is sulfate volatile? It could simply be because the fraction of nitrate decreased, causing that of sulfate to increase relatively. In addition to the effects of atmospheric dilution and dispersion, could the differences in dust composition from various sources also play a role?

**Reply:** Thank you very much for your insightful comment. Indeed, sulfate is considered a non-volatile species in $PM_{2.5}$. Therefore, volatilization effect was not a reason for the observed

increase in sulfate fraction during dust events. Rather, the relative increase could partially result from the decrease in nitrate fraction, as the referee suggested.

In addition, we agree that differences in dust composition originating from various source regions might also influence the observed sulfate levels. The increased fraction of sulfate during dust storms likely reflects a combination of meteorological effects and dust-related chemical processes. For instance, high concentrations of non-volatile cations (NVCs), such as $Ca^{2+}$, present during dust events can preferentially react with sulfate, forming stable species such as $CaSO_4$ instead of reacting with ammonia. These pathways have been discussed in earlier studies (e.g., Vasilakos et al., 2018; Guo et al., 2018). Furthermore, the entrainment of large amounts of mineral dust under high humidity conditions may enhance the heterogeneous oxidation of $SO_2$, particularly in urban environments with substantial $NO_x$ emissions, thereby facilitating sulfate formation (He et al., 2014).

Nitrate is generally formed more locally and is thus more sensitive to short-term meteorological dispersion and dilution, whereas sulfate typically exhibits a stronger regional signature. To further evaluate this hypothesis, we applied a random forest regression model coupled with SHAP analysis. Meteorological parameters (listed in Table S4) were used as input predictors, while nitrate and sulfate concentrations served as the respective response variables. As shown in Fig. S2, the aggregated SHAP values for meteorological variables associated with dispersion processes (e.g., winds, planetary boundary layer height) revealed a stronger influence on nitrate concentrations than on sulfate. This suggests that meteorological dilution has a more pronounced effect on nitrate, helping to explain its reduced relative abundance during dust storms. Meanwhile, the relatively stable SHAP responses for sulfate – and some increases during dust storm periods – may point to contributions from long-range transport of sulfate-containing particles. However, this interpretation would benefit from further verification through chemical transport modeling.

We have modified the manuscript which can be found on lines 241 – 260 in the revised manuscript.

**Table. S4** List of input meteorological parameters for simulating the effects of atmospheric dilution and dispersion on $SO_4^{2-}$ and $NO_3^-$ using a random forest model.

| Variable Abbreviation | Full Variable Name | Unit |
|---|---|---|
| U10 | 10m u-component of wind | m s$^{-1}$ |
| V10 | 10m v-component of wind | m s$^{-1}$ |
| U850 | 850hPa u-component of wind | m s$^{-1}$ |
| V850 | 850hPa v-component of wind | m s$^{-1}$ |
| W850 | 850hPa w-component of wind | m s$^{-1}$ |
| U650 | 650hPa u-component of wind | m s$^{-1}$ |
| V650 | 650hPa v-component of wind | m s$^{-1}$ |
| W650 | 650hPa w-component of wind | m s$^{-1}$ |
| U500 | 500hPa u-component of wind | m s$^{-1}$ |
| V500 | 500hPa v-component of wind | m s$^{-1}$ |
| W500 | 500hPa w-component of wind | m s$^{-1}$ |
| BLH | Boundary layer height | m |

[Figure]

**Fig. S2 Time series of the effects of atmospheric dilution and dispersion on NO$_3^-$ and SO$_4^{2-}$.** The y-axis represents the SHAP values of 12 factors simulated using a random forest model (detailed factors are in Table S4). Blue and red dots indicate the contributions of atmospheric dilution and dispersion effects to NO$_3^-$ and SO$_4^{2-}$, respectively.

3) Line 290: What scientific rationale underlies the specific concentration thresholds (0, 0.7, 3 ug m$^{-3}$) adopted for classification purposes?

**Reply:** Thank you for your insightful comment. The selected Ca$^{2+}$ concentration thresholds (0,

0.7, and 3 μg m$^{-3}$) represent three distinct atmospheric scenarios: non-dust, local dust, and extreme dust storm conditions, respectively. These thresholds were derived based on the variation trend of $Ca^{2+}$ concentrations observed in the sensitivity analysis shown in Figure 9 (Section 3.4). Specifically, 0 μg m$^{-3}$ was used to represent the baseline, non-dust scenario. A value of 0.7 μg m$^{-3}$ corresponds to the average $Ca^{2+}$ concentration during local dust events. The threshold of 3 μg m$^{-3}$, representing extreme dust storm conditions, was determined by calculating the mean $Ca^{2+}$ concentration during dust storm episodes plus one standard deviation. This approach ensures that the upper-end scenario captures elevated mineral dust loading. These three representative levels were used to explore the response of aerosol pH to changes in sulfate and ammonia under varying dust intensities, helping to isolate the role of mineral cations in buffering atmospheric acidity during dust events.

4) Lines 297-299: While the impact of $Ca^{2+}$ is demonstrated, were other cations (e.g., Fe and Mn) similarly evaluated?

**Reply:** Thank you for your valuable comment. In this study, aerosol pH was estimated using the ISORROPIA-II thermodynamic equilibrium model, which considers major inorganic species including total ammonium, $Na^+$, $K^+$, $Ca^{2+}$, $Mg^{2+}$, $SO_4^{2-}$, total nitrate, HCl, and chloride as input variables. Due to model limitations, transition metals such as Fe and Mn were not explicitly included in the simulations. We fully agree that Fe and Mn may also influence aerosol acidity, particularly under dust-influenced conditions. Previous studies have shown that transition metal ions (TMIs), such as $Fe^{3+}$ and $Mn^{2+}$, can be adsorbed onto the surface of dust particles and act as catalysts in the heterogeneous oxidation of $SO_2$, thereby promoting sulfate formation and indirectly influencing aerosol pH (Zhang et al., 2024). Additionally, recent findings suggest that Fe can dissolve in water-soluble Ca-nitrate coatings commonly formed on dust particles, and this dissolution is closely linked to aerosol acidity (Zhi et al., 2025). Observational data further indicate that Fe solubility under dust conditions tends to increase with aerosol acidity and decrease with particle size in the 0.32–5.6 μm range (Liu et al., 2022).

We appreciate the reviewer's suggestion, which provides meaningful insights for our future work. In subsequent studies, we could plan to incorporate atmospheric chemical transport models capable of accounting for the roles of TMIs, to more comprehensively assess their impacts on aerosol acidity in dust-laden environments.

5) Figure 6: In Figure 6b, there is no change in aerosol pH when the sulfate concentration is below approximately 5 µg m$^{-3}$. Could you please clarify the reason for this?

**Reply:** Thank you for the insightful comment. The original version of Figure 6 contained over one million data points, which significantly hindered visualization performance. To improve clarity and computational efficiency, we reduced the number of data points and re-ran the simulations. Due to adjustments in the color scale settings, there may be slight visual discrepancies between the revised figure and the specific case you mentioned. However, the phenomenon is also clearly illustrated in Figure 6c, which we refer to in our revised explanation.

In Figure 6c, When $SO_4^{2-}$ concentrations are low (below approximately 8 µg m$^{-3}$), the limited availability of $NH_4^+$ in the aerosol phase hampers the ability of $NH_3$ to fully neutralize acidic species, even under equilibrium conditions (Weber et al., 2016). Consequently, aerosol pH becomes less sensitive to changes in $SO_4^{2-}$ concentrations. This behavior is further modulated by the buffering role of $Ca^{2+}$. Under dust storm conditions, particularly in the extreme dust scenario, $Ca^{2+}$ is expected to preferentially react with $SO_4^{2-}$ before interacting with $NH_3$ (Vasilakos et al., 2018). When $SO_4^{2-}$ levels are low and $NH_3$ concentrations are held constant, $Ca^{2+}$ consumes much of the available $SO_4^{2-}$, thereby limiting the extent to which sulfate can regulate aerosol pH. This mechanism helps maintain relatively stable aerosol pH values under low sulfate conditions. The modification can be found on lines 316 - 321, in the revised manuscript.

6) Lines 357-358: Any reason? Is it due to the different composition of the dust or different environmental conditions?

**Reply:** Thank you for the comments. During dust storm events, the concentration of alkaline species such as $Ca^{2+}$ increased significantly — up to five times higher than during local dust events. This increase in $Ca^{2+}$ led to an elevated aerosol pH, which in turn significantly affected the partitioning of nitrate ($\varepsilon(NO_3^-)$). Following your question, we have added an explanation for the increase in $\varepsilon(NO_3^-)$, and the added content is as follows in Lines 382 -384: *"It is evident that the effect of dust storms on $\varepsilon(NO_3^-)$ is 10 to 20 times greater than the impact of local dust storm conditions, likely due to differences in aerosol composition and enhanced alkaline inputs such as $Ca^{2+}$."*

7) Lines 370-371: The interpretability of the random forest model's predictions requires further

clarification. For reductions ranging from 0% to 50%, do the concentrations of each species remain within the observed range after reduction? If not, the random forest model may fail to accurately capture the relationship between the predictors and the dependent variables, potentially leading to misinterpretations.

**Reply:** Thank you for your comments. We have addressed your concern in two parts: First, to ensure the validity of the sensitivity analysis, we carefully examined the pollutant concentrations associated with emission reductions ranging from 0% to 50%, as referred to in Line 398. We confirmed that, within this range of reduction, the concentrations of $TNH_x$, $TNO_3$, and $SO_4^{2-}$ remain within the bounds of observed values during the study period. A detailed summary table has been included in the supplementary material for your review. Second, we would like to clarify that the emission reduction analysis presented in Section 3.4 was conducted using the thermodynamic model ISORROPIA-II, rather than the random forest model. Therefore, the concerns regarding model extrapolation or interpretability of machine learning predictions do not apply in this context. The ISORROPIA-II model relies on established chemical thermodynamics, and its outputs reflect internally consistent equilibrium calculations based on input concentrations.

The revised description reads as follows: *"To explore the impact of emission reductions of $TNH_x$, $TNO_3$, and $SO_4^{2-}$ on $\varepsilon(NO_3^-)$ during different dust storm conditions, we conducted a sensitivity analysis based on the thermodynamic model ISORROPIA-II, using the average pollutant concentrations observed in Zhenjiang during the spring of 2023."* And *"Figure 9. Sensitivity analysis based on the thermodynamic model ISORROPIA-II simulated the impact of reducing $TNH_x$ ($TNH_x = NH_3 + NH_4^+$), $TNO_3$ ($TNO_3 = HNO_3 + NO_3^-$), and $SO_4^{2-}$ by 0-50% during dust events of varying intensities on $\varepsilon(NO_3^-)$ and $NH_4^+ + NO_3^-$."*

**Table 1.** The observed values and the range of changes in the emission reduction process data for $NH_4^+ + NO_3^-$, $SO_4^{2-}$, $TNH_x$, and $TNO_3$ in Zhenjiang based on sensitivity analysis.

|  | $NH_4^+ + NO_3^-$ | $SO_4^{2-}$ | $TNH_x$ | $TNO_3$ |
|---|---|---|---|---|
| Observation | 1.2-25.0 | 1.1-15.0 | 2.8-40.0 | 0.4-64.0 |
| Reduction 0-50% | 3.3-19.0 | 0.4-3.0 | 1.4-9.3 | 0.9-5.8 |

8) Figure 7: Please add into the figure typical pH values for non-dust periods for better clarity.

**Reply:** Thank you for the suggestion. We have added the pH values of non-dust periods. During non-dust periods, the average aerosol pH in all three cities is significantly lower than during

dust events, which helps to more clearly illustrate the impact of dust on ε(NO₃⁻). We have also added corresponding explanations in the main text Lines 339 – 342 : *"During non-dust periods, aerosol pH values in the three cities were significantly lower than during dust events (Xuzhou: 2.7 – 4.0, Zhenjiang: 2.2 – 3.7, Suzhou: 2.0 – 3.6). This lower pH corresponds to a marked decrease in ε(NO₃⁻), indicating a shift toward gaseous HNO₃, especially in Suzhou where ε(NO₃⁻) dropped to approximately 40% under the lowest pH conditions."* And the revised figure is shown below:

[Figure]

**Figure 7. S-curve distributions for ε(NO₃⁻) under the conditions from different cities.** Based on Eq. (2), the relationship between ε(NO₃⁻) and pH was calculated using the average T and $W_i$ during dust storm, local dust and non-dust periods (assuming $\gamma_{NO_3^-}\gamma_{H^+}$=0.28, $\gamma_{H^+}$ =1). The vertical dashed lines represent the minimum (left side) and maximum (right side) pH values under local-dust conditions calculated using ISORROPIA-II for the three cities. Error bars indicate the sample standard deviation of aerosol pH during local dust and dust storm events.

9) Lines 258-260: Add "for example" before this sentence to improve flow.

**Reply:** Thank you for the suggestion. We have added.

10) Please standardize all chemical notation with proper subscript/superscript formatting.

**Reply:** Thank you for the comments. We have checked all chemical symbol expressions in the manuscript and have used standardized subscript and superscript notation to ensure correct and consistent formatting.

**Reply to Anonymous Referee #2**

Comments: The manuscript explores the impact of dust pollution on aerosol pH and nitrate gas-particle partitioning. By combining field observations, thermodynamic modeling, and machine learning techniques, this study provides a comprehensive analysis of how different dust scenarios affect urban aerosol pH and gas-particle partitioning chemistry of nitrate. I would consider the publication of this article once the authors have addressed the following comments.

**Reply:** We thank the reviewer for his/her positive and constructive comments on this manuscript. We have modified the related content according to the comments.

1) (English) The language and grammar of the manuscript should be improved. In some of these instances, bad grammar prevents understanding the main point of the sentence. I will point in the minor comment section to these parts of the text, with a recommendation on how to change the text, based on my understanding of what the main point of it is.

**Reply:** Thank you for your valuable comments. We have carefully reviewed the language and grammar issues you highlighted and revised the corresponding sections to improve clarity and readability. These changes aim to ensure that the main points of the manuscript are conveyed more effectively. We sincerely appreciate your suggestions, which have helped enhance the overall quality of the manuscript.

2) L. 47: "Dust particles also engage in heterogeneous reactions with gaseous nitric acid, buffering acidic species and modulating pH dynamics." This is a priori reasonable description, due to the alkalinity of carbonate minerals. It would be good to have references here to support it. A recently published article in EST provides a thorough explanation of how aerosol acidification responds to the buffering capacity of carbonate minerals during Asian dust storms (https://pubs.acs.org/doi/10.1021/acs.est.4c12370).

**Reply:** Thank you for your insightful comment. We have carefully reviewed the suggested reference (https://pubs.acs.org/doi/10.1021/acs.est.4c12370), which indeed offers a comprehensive explanation of the buffering effects of carbonate minerals on aerosol acidification during Asian dust storms. As it directly supports and enriches our discussion on the role of dust in modulating aerosol pH, we have now cited this study at the appropriate place in the revised manuscript. We appreciate your recommendation, which has strengthened the scientific grounding of our work.

3) L. 75: "atmospheric setting" to "atmospheric experiment"

**Reply:** Thank you for the suggestion. We have revised "atmospheric setting" to "atmospheric experiment" as recommended to better reflect the context and improve clarity.

4) L. 231-236: The authors claim that the abundance of nitrate in aged dust particles during long-range transport dust storms was higher than during local dust periods, whereas sulfate abundance was greater during local dust periods than in long-range transport dust storms. You attribute this to the stronger atmospheric dilution and dispersion effects on nitrate during dust storms. You mean that long-range transport dilutes nitrate? If so, a decrease in nitrate contribution should be observed along the dust transport from Xuzhou to Zhenjiang to Suzhou. In reality, however, $Ca(NO_3)_2$ and $Mg(NO_3)_2$ coatings preferentially form on aged mineral particles containing calcite and dolomite. Moreover, the number of $Ca(NO_3)_2$-coated particles increases with dust transport distance due to the relatively low deliquescence relative humidities (>11%) (see Li et al., ACP, 2009, https://doi.org/10.5194/acp-9-1863-2009; Tobo et al., PNAS, 2010, http://www.pnas.org/cgi/doi/10.1073/pnas.1008235107; A. Laskin and T. W. Wietsma, JGR-A, 2005, doi:10.1029/2004JD005206). Given that calcite and dolomite are widely present in Asian dust particles, it is expected that during the dust storm period, the relative contribution of $Ca^{2+}$ and $Mg^{2+}$ increased across all three cities, with an average rise of approximately 10% compared to the local dust period. (your Figure 3). Therefore, the authors should thoroughly study more relevant literatures to provide a more reasonable explanation. Furthermore, the higher sulfate content in long-range transported dust particles likely originates from the presence of weakly soluble $CaSO_4$.

**Reply:** Thank you for this insightful comment and valuable references. We have revised the manuscript accordingly and cited the suggested studies to strengthen our discussion. As shown in Figure 3, the relative contribution of nitrate was indeed lower during dust storm periods compared to local dust events, whereas sulfate showed the opposite trend. This pattern was likely driven by the combined influence of meteorological conditions and dust-related chemical processes.

Specifically, nitrate, which is often formed locally through gas-particle conversion processes, is more sensitive to dispersion effects during dust storms than sulfate, which typically has stronger regional characteristics and may be less impacted by local meteorological

changes. To further evaluate this hypothesis, we applied a random forest regression model coupled with SHAP analysis. Meteorological parameters (listed in Table S4) were used as input predictors, while nitrate and sulfate concentrations served as the respective response variables. As shown in Fig. S2, the aggregated SHAP values for meteorological variables associated with dispersion processes (e.g., winds, planetary boundary layer height) revealed a stronger influence on nitrate concentrations than on sulfate. This suggests that meteorological dilution has a more pronounced effect on nitrate, helping to explain its reduced relative abundance during dust storms. Meanwhile, the relatively stable SHAP responses for sulfate – and some increases during dust storm periods – may point to contributions from long-range transport of sulfate-containing particles. However, this interpretation would benefit from further verification through chemical transport modeling.

In addition, we fully agree with the reviewer that chemical aging processes during dust transport likely contribute to nitrate formation. As reported in previous studies (Li and Shao, 2009; Tobo et al., 2010; Laskin et al., 2005), $Ca(NO_3)_2$ and $Mg(NO_3)_2$ coatings preferentially form on aged dust particles containing calcite and dolomite, with their abundance increasing along transport paths due to low deliquescence relative humidities. Given the widespread presence of these minerals in Asian dust, it is reasonable to expect enhanced formation of nitrate salts during long-range transport. Our measurements support this, as shown in Figure 3, where the relative contributions of $Ca^{2+}$ and $Mg^{2+}$ increased by approximately 10% during the dust storm period across all three cities compared to local dust events. This highlights the importance of both physical dispersion and chemical processing in shaping the observed aerosol composition. Nevertheless, further quantitative analysis of these factors requires comprehensive aerosol chemical measurements and atmospheric chemical transport modeling in future studies.

We have modified the manuscript which can be found on lines 241 – 260 in the revised manuscript.

5) L. 318: "During dust storms, the mean aerosol pH values were 5.50 ± 1.65 in Xuzhou, 5.44 ± 1.69 in Zhenjiang, and 5.30 ± 1.67 in Suzhou. Under local dust conditions, these values were lower, at 4.12 ± 0.52, 3.92 ± 0.32, and 3.74 ±0.69 respectively." As the author pointed out, dust particles were more acidified by more secondary acidic aerosols ($SO_4^{2-}$, $NO_3^-$) formed on dust surfaces along with dust long-distance transport, eventually leading to the decrease of pH.

However, why does a similar trend also appear during local dust events?

**Reply:** Thank you for the comment. The observed decrease in aerosol pH during local dust events across the three cities was primarily attributed to the reduction in ambient $NH_3$ concentrations. As shown in Table S1–S3, the average $NH_3$ concentrations during local dust events were $13.24 \pm 4.28$ µg m$^{-3}$ in Xuzhou, $9.27 \pm 3.99$ µg m$^{-3}$ in Zhenjiang, and $6.16 \pm 3.51$ µg m$^{-3}$ in Suzhou. As discussed in Section 3.2, $NH_3$ was identified as the most influential chemical species affecting aerosol pH. Therefore, the decrease in $NH_3$ concentrations during local dust conditions likely played a dominant role in driving the observed reduction in aerosol pH.

6) L. 359: "This indicates that dust storm conditions have a significantly stronger positive contribution to the particle-phase fraction of nitrate. The presence of dust particles facilitates the conversion of nitrate to the particulate phase, highlighting the significant influence of dust storms on nitrate partitioning in the atmosphere." Does this contradict the previous description of stronger atmospheric dilution and dispersion effects on nitrate? similar to Comment 5.

**Reply:** We thank the reviewer for the comment. The influence of dust storms on nitrate can indeed be understood from two distinct perspectives. In earlier sections, we focused on the physical processes of atmospheric dilution and dispersion, which primarily lead to a decrease in the absolute mass concentration of nitrate aerosols. In contrast, the discussion in Lines 384 – 387 centers on the thermodynamic gas-particle partitioning process. Specifically, in Section 3.3, we used machine learning methods to analyze how dust storms affect aerosol pH and the particle-phase fraction of nitrate ($\varepsilon(NO_3^-)$). While overall nitrate concentrations may decrease due to dispersion, the presence of alkaline mineral components (e.g., $Ca^{2+}$) in dust elevates aerosol pH and thereby promotes the partitioning of nitrate into the particle phase. Therefore, the observed increase in $\varepsilon(NO_3^-)$ reflects a relative enhancement in particle-phase nitrate formation and does not contradict the decline in total nitrate concentrations caused by atmospheric dispersion.

7) Figure 9: Please label the different dust pollution conditions along with $Ca^{2+}$ concentrations in the figure.

**Reply:** Thank you for the valuable suggestion. We have now labeled the different dust pollution

conditions along with the corresponding $Ca^{2+}$ concentrations in Figure 9a. To better represent the variation in calcium ion levels, we included $Ca^{2+}$ concentrations typical of local dust events (0.5 µg m$^{-3}$), and, using the ISORROPIA-II model, we simulated changes in $\varepsilon(NO_3^-)$ and $NH_4^+$ + $NO_3^-$ under three emission reduction scenarios. We have updated the manuscript as follows: "*For the simulation, $Ca^{2+}$ concentration was set to 0.1 to 0.7 µg m$^{-3}$ for local dust conditions and ranged from 1.0 to 3.0 µg m$^{-3}$ for dust storm conditions.*" The revised figure is shown below:

[Figure]

**Figure 9.** Sensitivity analysis based on the thermodynamic model ISORROPIA-II simulated the impact of reducing $TNH_x$ ($TNH_x$ = $NH_3$ + $NH_4^+$), $TNO_3$ ($TNO_3$ = $HNO_3$ + $NO_3^-$), and $SO_4^{2-}$ by 0-50% during dust events of varying intensities on $\varepsilon(NO_3^-)$ and $NH_4^+$ + $NO_3^-$.

**References**

Guo, H., Nenes, A., and Weber, R. J.: The underappreciated role of nonvolatile cations in aerosol ammonium-sulfate molar ratios, Atmos. Chem. Phys., 18, 17307-17323, https://doi.org/10.5194/acp-2017-737, 2018.

He, H., Wang, Y., Ma, Q., Ma, J., Chu, B., Ji, D., Tang, G., Liu, C., Zhang, H., and Hao, J.: Mineral dust and NOx promote the conversion of SO2 to sulfate in heavy pollution days, Sci. Rep., 4, 4172, https://doi.org/10.1038/srep04172, 2014.

Laskin, A., Wietsma, T. W., Krueger, B. J., and Grassian, V. H.: Heterogeneous chemistry of individual mineral dust particles with nitric acid: A combined CCSEM/EDX, ESEM, and ICP-MS study, J. Geophys. Res. Atmos., 110, https://doi.org/10.1029/2004JD005206, 2005.

Li, W., Qi, Y., Liu, Y., Wu, G., Zhang, Y., Shi, J., Qu, W., Sheng, L., Wang, W., Zhang, D., and Zhou, Y.: Daytime

and nighttime aerosol soluble iron formation in clean and slightly polluted moist air in a coastal city in eastern China, Atmos. Chem. Phys., 24, 6495-6508, https://doi.org/10.5194/acp-24-6495-2024, 2024.

Li, W. J. and Shao, L. Y.: Observation of nitrate coatings on atmospheric mineral dust particles, Atmos. Chem. Phys., 9, 1863-1871, https://doi.org/10.5194/acp-9-1863-2009, 2009.

Liu, L., Li, W., Lin, Q., Wang, Y., Zhang, J., Zhu, Y., Yuan, Q., Zhou, S., Zhang, D., Baldo, C., and Shi, Z.: Size-dependent aerosol iron solubility in an urban atmosphere, npj Clim. Atmos. Sci., 5, 53, https://doi.org/10.1038/s41612-022-00277-z, 2022.

Rumsey, I. C., Cowen, K. A., Walker, J. T., Kelly, T. J., Hanft, E. A., Mishoe, K., Rogers, C., Proost, R., Beachley, G. M., Lear, G., Frelink, T., and Otjes, R. P.: An assessment of the performance of the Monitor for AeRosols and GAses in ambient air (MARGA): a semi-continuous method for soluble compounds, Atmos. Chem. Phys., 14, 5639-5658, https://doi.org/10.5194/acp-14-5639-2014, 2014.

Tobo, Y., Zhang, D., Matsuki, A., and Iwasaka, Y.: Asian dust particles converted into aqueous droplets under remote marine atmospheric conditions, Proc. Natl. Acad. Sci. U.S.A., 107, 17905-17910, https://doi.org/10.1073/pnas.1008235107, 2010.

Trebs, I., Meixner, F. X., Slanina, J., Otjes, R., Jongejan, P., and Andreae, M. O.: Real-time measurements of ammonia, acidic trace gases and water-soluble inorganic aerosol species at a rural site in the Amazon Basin, Atmos. Chem. Phys., 4, 967-987, https://doi.org/10.5194/acp-4-967-2004, 2004.

Vasilakos, P., Russell, A., Weber, R., and Nenes, A.: Understanding nitrate formation in a world with less sulfate, Atmos. Chem. Phys., 18, 12765-12775, https://doi.org/10.5194/acp-18-12765-2018, 2018.

Wang, T., Liu, Y., Cheng, H., Wang, Z., Fu, H., Chen, J., and Zhang, L.: Significant formation of sulfate aerosols contributed by the heterogeneous drivers of dust surface, Atmos. Chem. Phys., 22, 13467-13493, https://doi.org/10.5194/acp-22-13467-2022, 2022.

Weber, R. J., Guo, H., Russell, A. G., and Nenes, A.: High aerosol acidity despite declining atmospheric sulfate concentrations over the past 15 years, Nat. Geosci., 9, 282-285, https://doi.org/10.1038/ngeo2665, 2016.

Zhang, H., Li, R., Dong, S., Wang, F., Zhu, Y., Meng, H., Huang, C., Ren, Y., Wang, X., Hu, X., Li, T., Peng, C., Zhang, G., Xue, L., Wang, X., and Tang, M.: Abundance and Fractional Solubility of Aerosol Iron During Winter at a Coastal City in Northern China: Similarities and Contrasts Between Fine and Coarse Particles, J. Geophys. Res. Atmos., 127, e2021JD036070, https://doi.org/10.1029/2021JD036070, 2022.

Zhang, S., Li, D., Ge, S., Wu, C., Xu, X., Liu, X., Li, R., Zhang, F., and Wang, G.: Elucidating the Mechanism on the Transition-Metal-Ion-Synergetic-Catalyzed Oxidation of SO2 with Implications for Sulfate Formation in Beijing Haze, Environ. Sci. Technol., 58, 2912-2921, https://doi.org/10.1021/acs.est.3c08411, 2024.

Zhi, M., Wang, G., Xu, L., Li, K., Nie, W., Niu, H., Shao, L., Liu, Z., Yi, Z., Wang, Y., Shi, Z., Ito, A., Zhai, S., and Li, W.: How Acid Iron Dissolution in Aged Dust Particles Responds to the Buffering Capacity of Carbonate Minerals during Asian Dust Storms, Environ. Sci. Technol., https://doi.org/10.1021/acs.est.4c12370, 2025.